# Study on Multiphase Flow in a Wide-Width Continuous Casting Mold

**Lei Ren [1,\*]**, **Wenxiang Liu [1]**, **Haitao Ling [2,\*]** and **Jichun Yang [1]**

1 School of Materials and Metallurgy (Rare Earth School), Inner Mongolia University of Science and Technology (IMUST), Baotou 014010, China; liu1652447316@163.com (W.L.); yangjichun1963@163.com (J.Y.)
2 School of Metallurgical Engineering, Anhui University of Technology, Ma'anshan 243002, China
\* Correspondence: renlei19890817@163.com (L.R.); linghaitao@ahut.edu.cn (H.L.)

**Abstract:** The multiphase flow in the mold has a significant impact on the surface quality of the slab. In this paper, the multiphase flow in the mold is studied by establishing a one-quarter scale water mold, with the aid of a high-speed camera and particle image velocimetry (PIV). The oil phase will make the liquid surface velocity around the nozzle smaller. The greater the viscosity of the oil, the greater the critical water model casting speed and the shallower the critical immersion depth of submerged entry nozzle (SEN). Blowing will enhance the turbulence of the flow field in the mold and have a suppressing effect on the surface velocity. However, the vertical velocity of the narrow surface does not change significantly. The randomness of the bubble entering the mold from the nozzle can easily cause asymmetry of the instantaneous flow. The number of bubbles with a diameter less than 1 mm increase with the increase in gas flow rate. The larger the bubble size, the more buoyant around the nozzle when it escapes. The larger the diameter of bubble, the closer the vortex center of the upper circulation is to the nozzle and the closer the center of the lower circulation is to the narrow surface.

**Keywords:** multiphase flow; wide-width; CC mold; fluid flow

## 1. Introduction

The flow of molten steel in a mold is a complex multiphase fluid flow; according to statistics, 80% of billet defects come from the mold. In continuous casting, slag entrapment may occur due to turbulence at the interface between steel and slag [1,2]. Slag entrapment can create surface and internal defects that can damage the mechanical properties of the cast billet. Argon is blown into the mold to prevent clogging of the nozzle and promote the inclusion floatation from the molten steel [3]. However, the bubbles carried in the jets of molten steel will greatly affect the flow pattern in the mold, which will affect the flow behavior of molten steel.

As for the slag phase, Wang et al. [4] believed that the flow behavior of liquid mold slag was greatly affected by the fluctuation of liquid level. The discontinuous flow of mold slag leads to the formation of a solid slag film with uneven thickness, and the irregular solid film in turn obstructs the inflow of liquid mold slag. Liu et al. [5] found through physical simulation that only when the surface flow rate of molten steel is greater than the critical flow rate of slag entrapment, and when the immersion depth of mold slag exceeds the critical depth, it may produce mold slag defects. Iguchi et al. [6] studied the effect of viscosity of mold slag on slag entrapment. Zhang et al. [7] found that the important factor affecting the absorption of slag and inclusions is the surface fluctuation of the velocity, which is caused by the flow field distribution inside the mold. Bielnicki et al. [8] found that the conditions of entrained oil are variable and depend on experimental factors. For oils with viscosities of 0.0528 and 0.3748 Pa·s, the presence of mold powder influences the increase in critical water model casting speed. For oils with a viscosity of 0.1205 Pa·s, the mold powder has the opposite effect.

As for the gas–liquid phase, Deng et al. pointed out that the argon blowing flowrate is one of the decisive factors affecting molten steel flow pattern in molds. In order to maintain the double strand flow pattern in a mold, the argon blowing flowrates in a continuous casting process are reduced to be lower than the critical argon blowing flowrates in actual casting. Yin et al. [9] believed that argon blowing would weaken the flow in the upper circulation region and reduce the downward angle of the molten steel jet. Javurek et al. [10] studied the influence of bubbles on the flow of fluid in the mold through numerical simulation, and believed that turbulence would affect the rising speed of bubbles. Chen et al. [11] found that gas injection can directly affect the flow field and bubble behavior. With the increase in gas flow, the impact point on the narrow surface and the vortex core in the upper circulation region move upward. Liu et al. [5] established the Euler–Euler Large Eddy Simulation (EELES) method for the two-phase flow of argon and steel in slab continuous casting molds. Two typical transient flow structures were found in the upper rolls, consisting of vortices with clockwise and counterclockwise rotation directions, respectively. Large bubbles rise toward the top surface quickly, while smaller bubbles are carried deep into the mold.

Table 1 shows a large number of previous studies conducted on multiphase flow in molds. In this paper, the wide-width mold (prototype width of 2040 mm) was taken as the research object, using a high-speed camera and Particle Image Velocimetry. Firstly, the critical conditions of slag coiling with different viscosity of mold slag were studied. Secondly, the bubble size distribution and gas–liquid two-phase flow mechanism in the mold were analyzed, which provides a reference for improving slab quality.

**Table 1.** Previous studies on multiphase flow.

| Year | Authors | Main Conclusion | References |
|------|---------|-----------------|------------|
| 1998 | Bouris et al. | Slag entrapment may occur due to turbulence at the interface between steel and slag. | [1,2] |
| 2000 | Iguchi et al. | Studied the effect of kinematic viscosity of mold slag on slag entrapment. | [6] |
| 2006 | Zhang et al. | Blowing promotes the inclusion floatation from the molten steel. | [3] |
| 2013 | Liu et al. | Established the Euler–Euler Large Eddy Simulation (EELES) method for the two-phase flow of argon and steel in slab continuous casting molds. | [5] |
| 2018 | Bielnicki et al. | Laboratory experiments of mold slag entrainment in the continuous casting process were carried out. | [8] |
| 2019 | Zhang et al. | The risk of surface slag entrainment was discussed from the perspectives of flow patterns, surface fluctuation, and velocity. | [7] |
| 2019 | Chen et al. | Gas injection can directly affect the flow field and bubble behavior. | [11] |
| 2020 | Zhang et al. | Argon blowing would weaken the flow in the upper circulation region. | [9] |
| 2020 | Javurek et al. | Turbulence would affect the rising speed of bubbles. | [10] |

## 2. Experiment

### 2.1. Water Model Experiments

A one-quarter scale water model of the slab mold made of transparent plastic was built with the dimensions shown in Figure 1, which satisfies the condition that Froude numbers are equal in the similarity principle. Figure 1 shows the structure and size of the mold. The water flow from the nozzle down into the mold is controlled through a stopper rod. A metering pump is used for water recycling, with a water model casting speed of 0.425 m·min$^{-1}$. The submergence depth of the SEN, from the top exit port to the meniscus, is 40 mm. Water discharges from the bottom of the water model of mold through a perforated plastic plate arranged $4 \times 42$, which is used for weakening the bottom vortexing flow caused by drainage pipes (8.8 mm in diameter). Then, water discharges the model through three outlets. Then, water passes through a magnified diameter pipe, and is pumped back up to the tundish through a flow meter. The experimental apparatus is shown in Figure 2. The water model casting speed of the actual mold is 0.85 m·min$^{-1}$, after the derivation of Equations (1) to (3), the ratio of the water model casting speed of the mold to that of the prototype can be calculated as 0.5, and the water model casting speed of the

mold is 0.425 m·min$^{-1}$. A two-port SEN was built to feed water into the plastic mold, as shown in Figure 3. Table 2 shows the details of dimensions and casting conditions of the water model and the corresponding full-scale steel caster.

$$Fr_m = Fr_p \tag{1}$$

$$\frac{V_m^2}{gl_m} = \frac{V_p^2}{gl_p} \Rightarrow \frac{V_m^2}{l_m} = \frac{V_p^2}{l_p} \tag{2}$$

$$\frac{V_m}{V_p} = \sqrt{\frac{l_m}{l_p}} = \sqrt{\frac{1}{4}} = \frac{1}{2} = 0.5 \tag{3}$$

where the subscript $m$ represents the physical quantity corresponding to the water model and $p$ represents the physical quantity corresponding to the prototype.

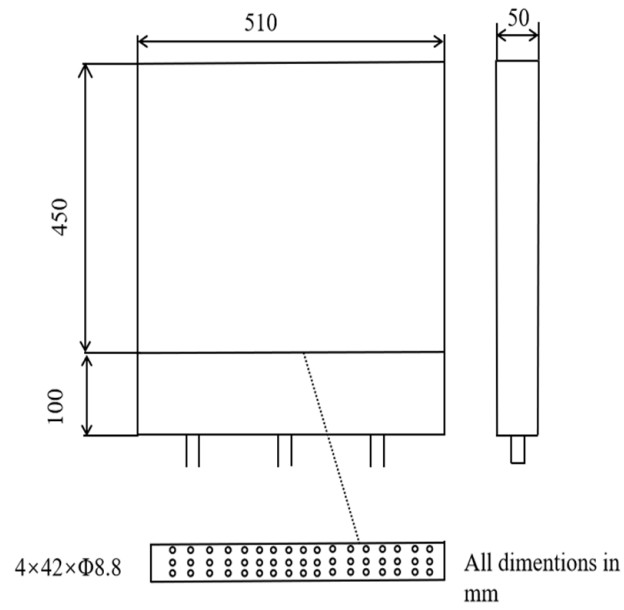

**Figure 1.** Structure and size of the water model.

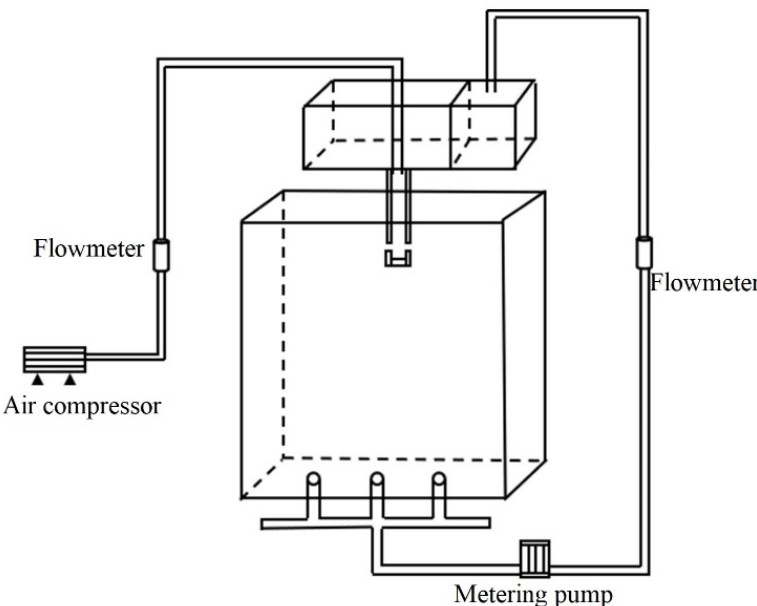

**Figure 2.** Water model experimental apparatus.

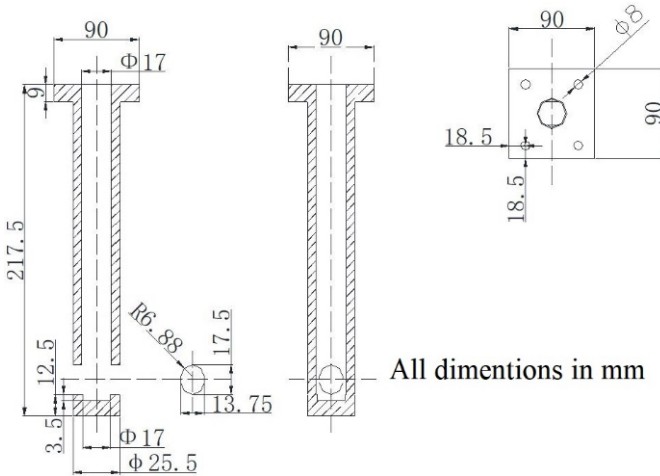

**Figure 3.** Dimensions and geometries of SEN.

**Table 2.** Parameters of the water model and the corresponding full-scale steel caster.

| Parameter | Actual Mold | 0.25 Scale Water Model |
|---|---|---|
| Mold section | $2040 \times 200$ mm | $510 \times 50$ mm |
| Cast speed | $0.85$ m·min$^{-1}$ | $0.425$ m·min$^{-1}$ |
| Nozzle diameter | 70 mm | 17.5 mm |
| Flow rate in nozzle | $1.50$ m·s$^{-1}$ | $0.83$ m·s$^{-1}$ |
| Nozzle angle | $0°$ | $0°$ |
| Viscosity | $5.23 \times 10^{-3}$ Pa·s (1490 °C) | $1.00 \times 10^{-3}$ Pa·s (20 °C) |
| Density | $6932$ kg·m$^{-3}$ (1490 °C) | $998$ kg·m$^{-3}$ (20 °C) |

Silicone oil was used in the experiments to simulate the slag that covers the liquid surface of the mold in actual production. Table 3 shows the physical properties of steel and slag in actual continuous casting, which correspond to the physical properties of water at room temperature and silicone oil.

**Table 3.** Physical properties of each liquid phase.

| Phase | $\rho$ (kg·m$^{-3}$) | $\mu$ ($10^{-3}$ Pa·s) | $\upsilon$ ($10^{-6}$ m$^2$·s$^{-1}$) |
|---|---|---|---|
| Oil A | 956 | 47.8 | 50 |
| Oil B | 956 | 95.6 | 100 |
| Oil C | 956 | 191.2 | 200 |
| Water (20 °C) | 998 | 1.00 | 1.00 |
| Slag (1490 °C) | 2890 | 110 | 38.06 |
| 304molten steel (1490 °C) | 6932 | 5.23 | 0.75 |

Since the slag and steel are mainly affected by the viscous force, only the ratio of the kinematic viscosity between slag and steel and the ratio of the kinematic viscosity between silicon oil and water are considered when selecting silicone oil, as shown in Equation (4). It can be calculated that the kinetic viscosity of the selected silicone oil should be $50.75 \times 10^{-6}$ m$^2$·s$^{-1}$, so the silicone oil A in Table 3 was used in this experiment.

$$v_{slag}/v_{steel} = v_{oil}/v_{water} \tag{4}$$

where $v_{slag}$ is the kinematic viscosity of the actual liquid slag used at the casting temperature (m$^2$·s$^{-1}$); $v_{steel}$ is the kinematic viscosity of the actual steel at the casting temperature (m$^2$·s$^{-1}$); $v_{oil}$ is the kinematic viscosity of the selected silicone oil at room temperature (m$^2$·s$^{-1}$); and $v_{water}$ is the kinematic viscosity of water at room temperature (m$^2$·s$^{-1}$).

### 2.2. Experimental Equipment

High-speed camera and PIV technology were used in this experiment. Figure 4 shows the high-speed camera used for the experiment, whose model is HS5C8GB. In this experiment, the frame size of the high-speed camera was set as $1616 \times 996$, the frame rate was set as 800 FPS, and the exposure time was set as 1570 μs. Figure 5 shows the working principle and equipment composition of PIV. See the literature for details [12].

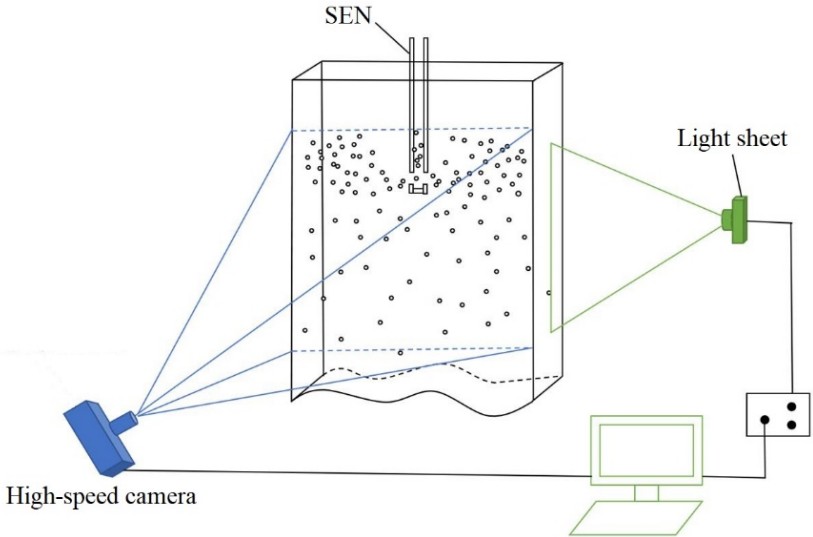

**Figure 4.** High-speed camera principle.

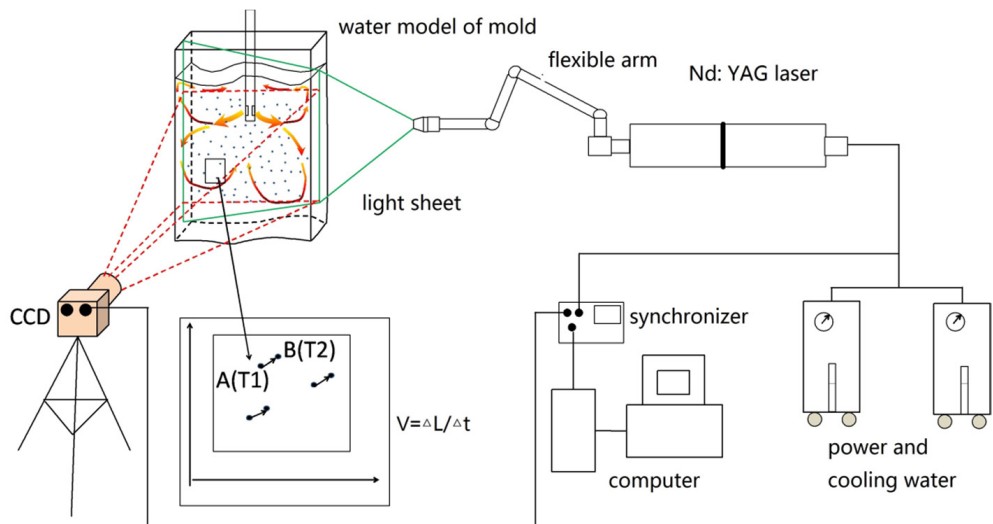

**Figure 5.** PIV principle.

## 3. Results and Discussion

### 3.1. Influence of Oil on Flow in a Wide Mold

Figure 6 shows the influence adding oil to the liquid level on the flow field in a wide slab casting mold. By comparing Figure 6a,b, it can be found that the addition of silicone oil mainly affects the liquid level flow rate of the mold, with little influence on the internal flow. Because of the viscous force between oil and water, the kinetic energy loss of the flow from the narrow surface to the nozzle is bigger, which could result in a small liquid surface velocity around the nozzle.

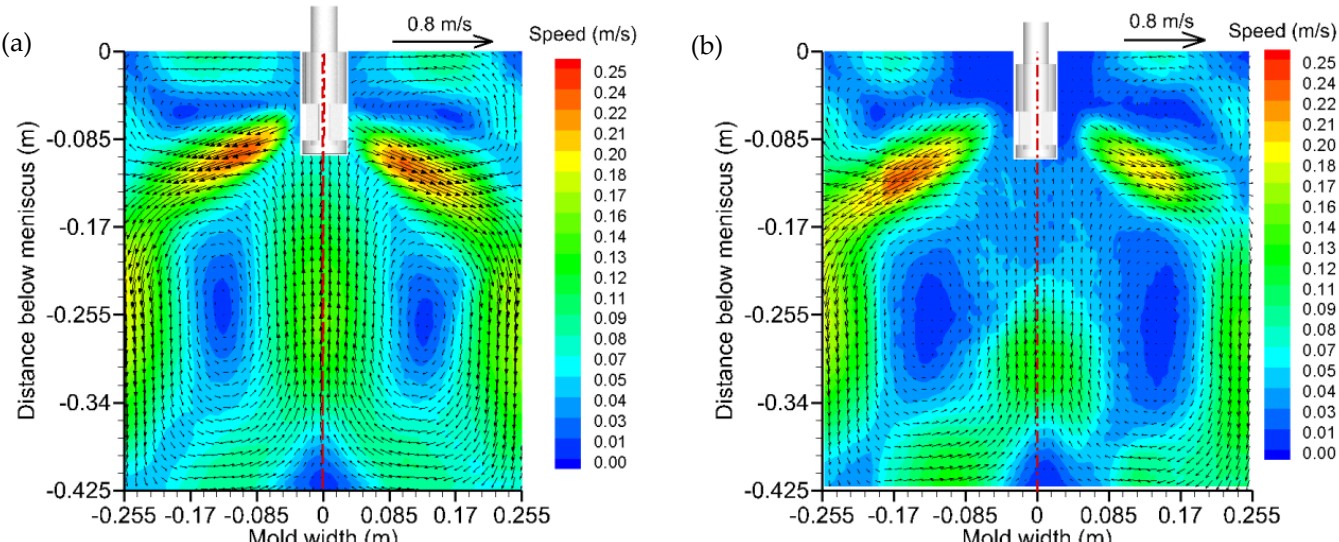

**Figure 6.** The influence of oil added to the liquid level on the flow field in the mold: (**a**) no oil; (**b**) oil thickness 3 mm.

Figure 7 shows the comparison of the surface horizontal velocity before and after adding the oil phase. Without adding silicone oil, the maximum horizontal velocity of the liquid level on the left side of the water outlet is 0.091 m·s$^{-1}$, and the velocity is still 0.02 m·s$^{-1}$ even near the outer wall of the water outlet. After adding silicone oil, at 85 mm from the nozzle, the maximum horizontal velocity on the left side of the outlet is reduced to 0.065 m·s$^{-1}$, the level velocity decreases to a very weak level. The right side of SEN is the same as the left side: without adding silicone oil, the velocity of the liquid level is 0.093 m·s$^{-1}$. At the position near the outer wall of the nozzle, the liquid level velocity is still 0.02 m·s$^{-1}$. After adding silicone oil, the maximum surface horizontal velocity of the liquid level on the right side of the nozzle decreases to 0.066 m·s$^{-1}$ and the surface velocity decreases to a very weak level at about 50 mm from the nozzle.

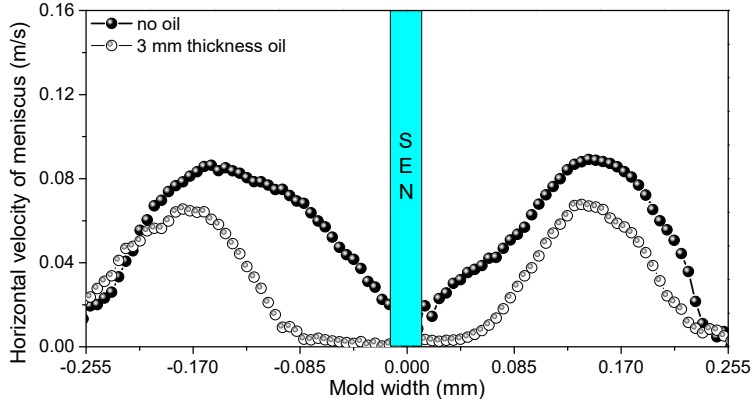

**Figure 7.** The influence of oil added to the liquid level on the horizontal velocity of the liquid level of the mold.

### 3.2. Critical Condition of Slag Entrapment at Liquid Level in Wide Slab Continuous Casting Mold

A high-speed camera was used to record the silicone oil droplets coming out of the oil phase into the interior of the mold. The results show that the narrow meniscus surface is exposed to upwelling impact, and the oil drops are drawn into the mold in the thick part of the oil layer. Figure 8 shows the uneven oil layer thickness in the width direction of the mold exposed locally on the narrow meniscus surface recorded by the high-speed camera.

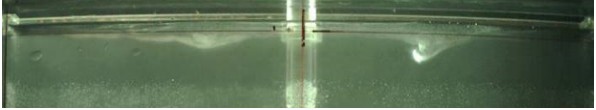 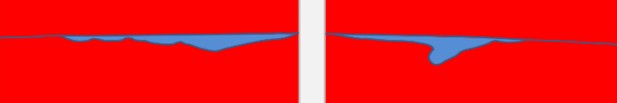

**Figure 8.** Oil entrapment photos.

The critical water model casting speed of silicone oil forming oil droplets and getting involved in water was studied using a high-speed camera as shown in Figure 9. The nozzle angle was 0°, the immersion depth was 50 mm, the silicone oil was selected as oil A, and the water model casting speed was gradually increased under these conditions. When the water model casting speed was 0.425 m·min$^{-1}$, as shown in Figure 9a, because of the upper circulation, part of the silicone oil at the narrow surface is pushed to about 1/8 of the mold. When the water model casting speed increases to 0.50 m·min$^{-1}$, as shown in Figure 9b, the upper circulation velocity becomes larger, the silicone oil at the narrow surface is pushed to about 1/4 of the mold, and liquid steel exposure begins to appear near the narrow surface. When the water model casting speed is increased to 0.55 m·min$^{-1}$, the upper circulation is further enhanced and the liquid surface velocity is accelerated, and the oil droplets may enter the mold at any time. When the water model casting speed increases to 0.60 m·min$^{-1}$, the phenomenon of oil droplets entering the mold for the first time occurs. Therefore, the critical water model casting speed under this condition is 0.60 m·min$^{-1}$. Using silicon oils B and C in the same way, the critical water model casting speeds of oil droplets into water are 0.62 and 0.65 m·min$^{-1}$, respectively. Taking viscosity as the abscissa and the corresponding critical water model casting speed as the vertical coordinate, the approximate relationship between the viscosity of silicone oil and the critical water model casting speed can be obtained when the nozzle immersion depth is 50 mm. It can be found that the greater the viscosity is, the greater the critical water model casting speed is, as shown in Figure 10a. The critical water model casting speed corresponding to different viscosity of mold slag can be found through this figure.

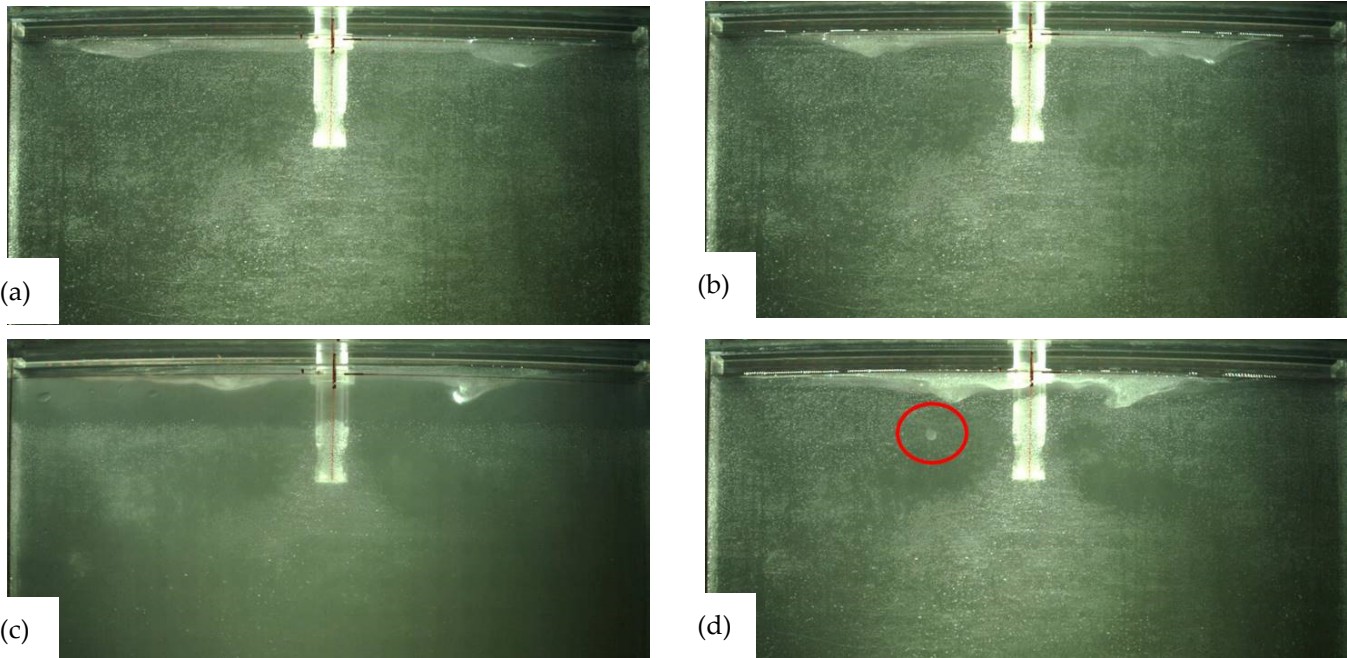

**Figure 9.** Water–oil interface phenomenon under different water model casting speed conditions (**a**) 0.425 m·min$^{-1}$, (**b**) 0.50 m·min$^{-1}$, (**c**) 0.55 m·min$^{-1}$, (**d**) 0.60 m·min$^{-1}$.

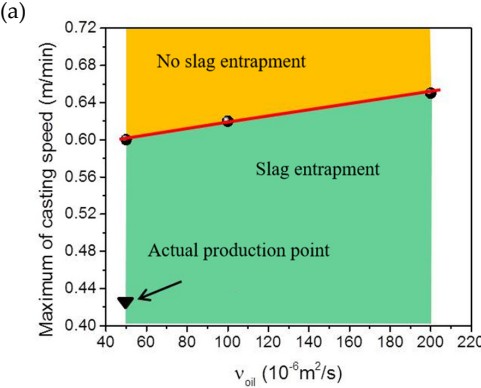
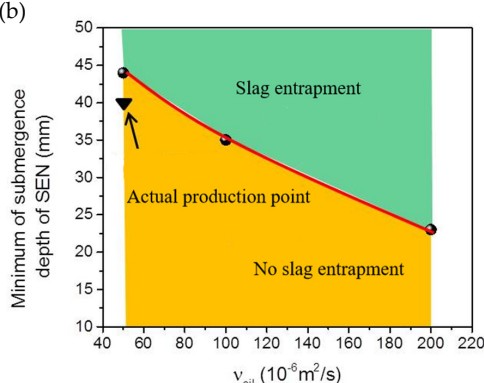

**Figure 10.** Critical conditions of oil droplet involvement. (**a**) The critical water model casting speed of oil droplet involvement when the nozzle immersion depth is 50 mm. (**b**) The critical nozzle immersion depth of oil droplets when the water model casting speed is 0.425 m·min$^{-1}$.

We also found the critical immersion depth using a high-speed camera. When the water model casting speed is 0.425 m·min$^{-1}$ and the silicone oils are A, B, and C, the critical immersion depth of the oil droplets from the oil phase into the mold is 44, 35, and 23 mm, respectively. The greater the viscosity, the shallower the critical immersion depth. Taking viscosity as the x-coordinate and the corresponding critical immersion depth as the y-coordinate, the approximate relationship between the viscosity of silicone oil and the critical immersion depth is obtained when the water model casting speed is 0.425 m·min$^{-1}$. The greater the viscosity, the shallower the critical immersion depth, as shown in Figure 10b. According to this figure, the critical immersion depth corresponding to different viscosity of slag can be found to prevent slag entrainment.

Table 4 shows the critical water model casting speed corresponding to the three silicone oils when the immersion depth is 50 mm, and the critical immersion depths corresponding to the three kinds of silicone oil when the water model casting speed is 0.425 m·min$^{-1}$.

**Table 4.** Critical condition of oil droplets with different silicone oil viscosity.

| Condition | Variable | Oil Drops Fall to the Critical Water Model Casting Speed and the Critical Nozzle Immersion Depth | | |
| --- | --- | --- | --- | --- |
| | | Oil A | Oil B | Oil C |
| Nozzle immersion depth 50 mm | Cast speed | 0.6 m·min$^{-1}$ | 0.62 m·min$^{-1}$ | 0.65 m·min$^{-1}$ |
| Cast speed 0.425 m·min$^{-1}$ | Immersion depth | 44 mm | 35 mm | 23 mm |

### 3.3. Study on the Behavior of Bubbles in a Wide Mold

Singh [13] found that, other than affecting the heat transfer and fluid flow in the mold, the gas bubble movement also dictates the quality of steel, as it is responsible for inclusion and bubble entrapment. Many scholars have used numerical simulations to study the behavior of the trajectory, morphological changes, and size distribution of bubbles in the mold after being ejected from the nozzle, etc. The size distribution of bubbles and its transient variation were measured using a high-speed camera, and by using the image processing software ImageJ for bubble size distribution in the image and statistics. Figure 11 shows the trajectories of bubbles.

Figure 12 shows the bubble size distribution in the mold at a certain moment under different blowing conditions. Due to the impact at the bottom of the nozzle and turbulent disturbance, some large sized bubbles break into small bubbles of different diameters. When the gas–liquid two-phase flow exits from the nozzle, the sizes of bubbles are different: the large diameter is 6 mm and the smallest diameter is just a fraction of a millimeter. As the large sized bubble is subjected to a large buoyancy force, it will soon float up to the liquid level and escape from the nozzle after injection, while the small sized bubble can be

carried to the deeper position of the mold by the flow stream [4,14]. It was found in the experiment that the bubbles with a diameter less than 1 mm have difficulty escaping from the mold, which agrees with previous results [15].

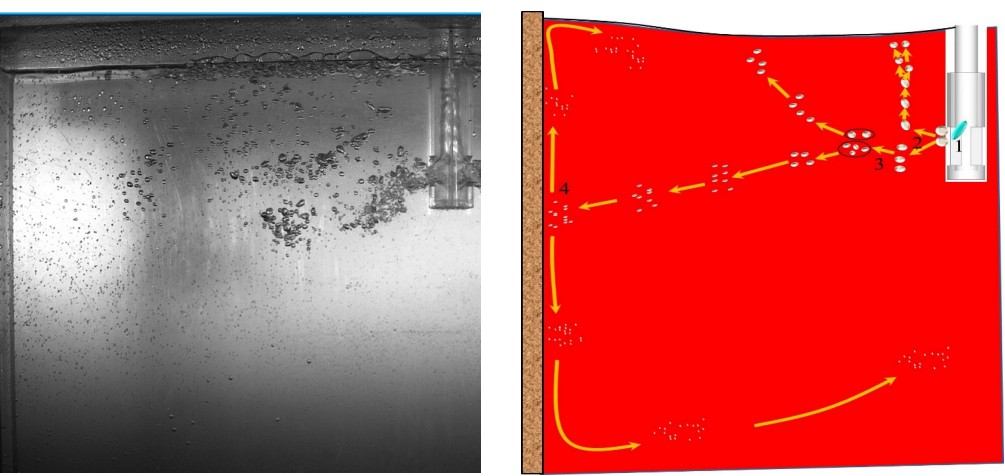

**Figure 11.** Bubble movement distribution.

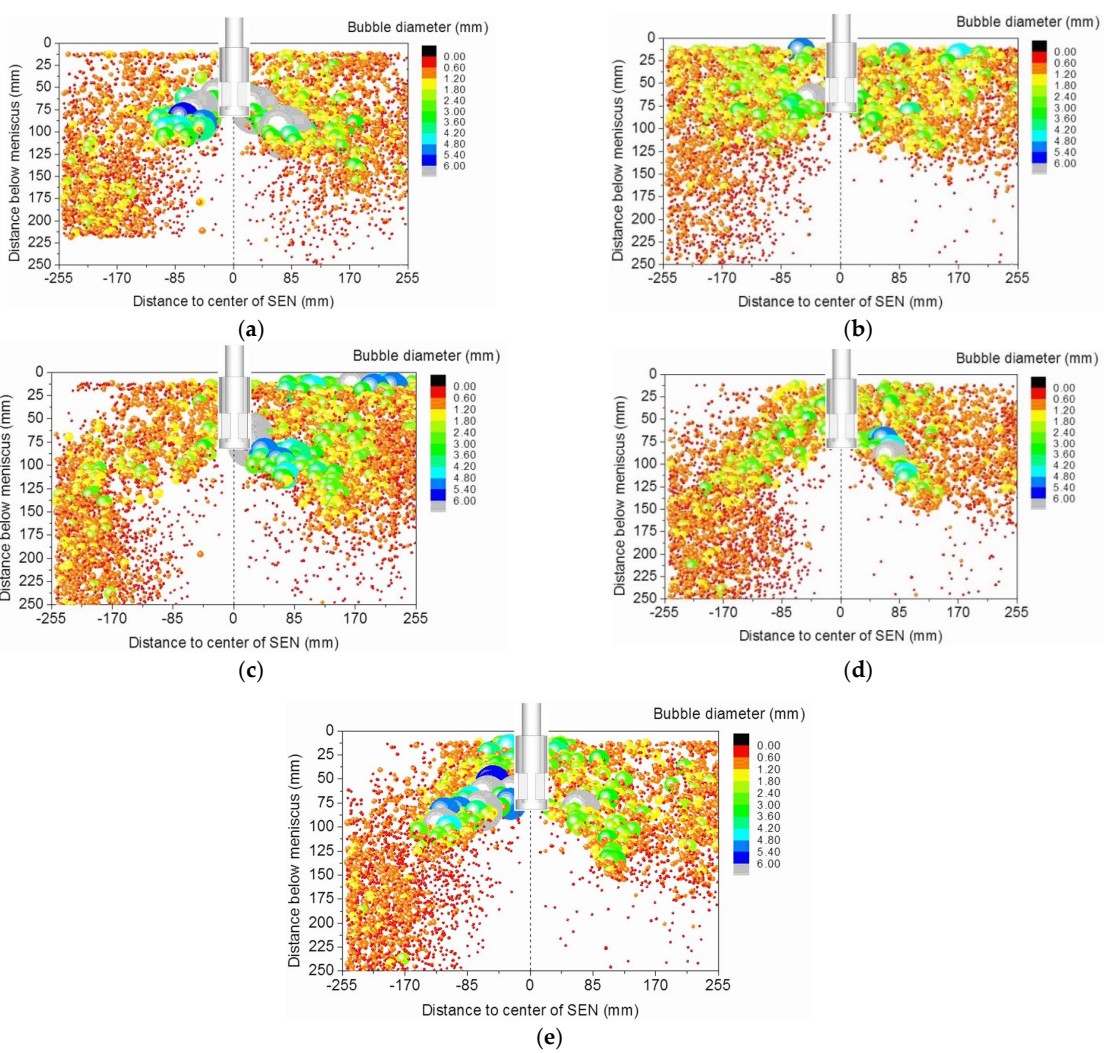

**Figure 12.** Distribution of bubbles in the mold: (**a**) 30 mL·min$^{-1}$, (**b**) 60 mL·min$^{-1}$, (**c**) 90 mL·min$^{-1}$, (**d**) 160 mL·min$^{-1}$, (**e**) 190 mL·min$^{-1}$.

Figure 13 shows the statistics of bubbles of different sizes on the left and right sides of the center line of the mold under different gas flow rates. It can be seen that when the gas flow rate is 30 mL·min$^{-1}$, the bubbles of different sizes on the left and right sides of the mold are evenly distributed. Most of the bubbles with a diameter less than 1 mm start to be unevenly distributed with the increase in the gas flow rate. It can be found that blowing is also one of the reasons for the asymmetric flow in the mold. With the increase in gas flow rate, the total number of bubbles in the mold tends to increase, mainly because the number of bubbles with a diameter of less than 1 mm increases.

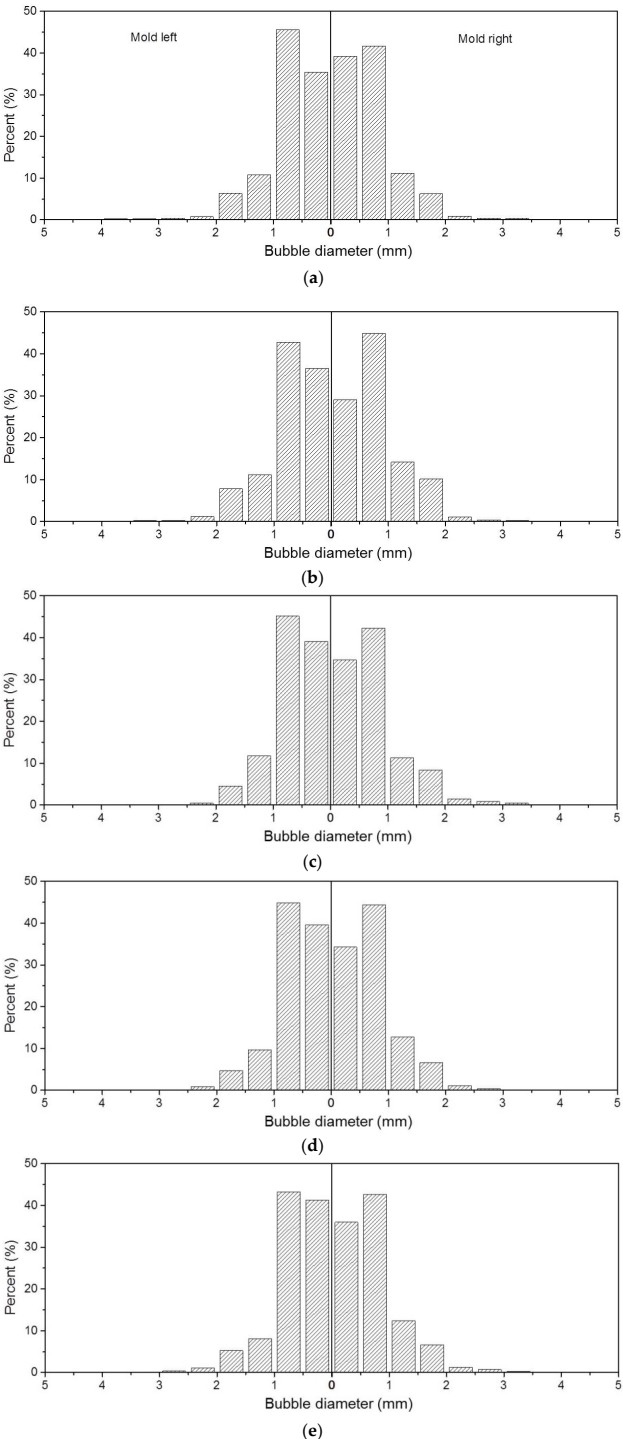

**Figure 13.** Size distribution of bubbles along the wide face of the mold: (**a**) 30 mL·min$^{-1}$, (**b**) 60 mL·min$^{-1}$, (**c**) 90 mL·min$^{-1}$, (**d**) 160 mL·min$^{-1}$, (**e**) 190 mL·min$^{-1}$.

Figure 14 shows the relationship between the gas flow rate and the average size of all bubbles in the mold. Its calculation formula is shown in (5). When the gas flow rates are 30, 60, 90, 160, and 190 mL·min$^{-1}$, respectively, the average diameters of all bubbles in the mold are 0.69, 0.76, 0.71, 0.69, and 0.71 mm, respectively. It can be found that appropriately increasing the gas flow rate can increase the bubble size, but not always.

$$d_{ave} = \sqrt[3]{\frac{\sum_{i=1}^{N} d_i{}^3}{N}} \tag{5}$$

where $d_{ave}$ is the average diameter of the bubble (mm); $N$ is the number of bubbles; and $D_i$ is the diameter of the bubble (mm).

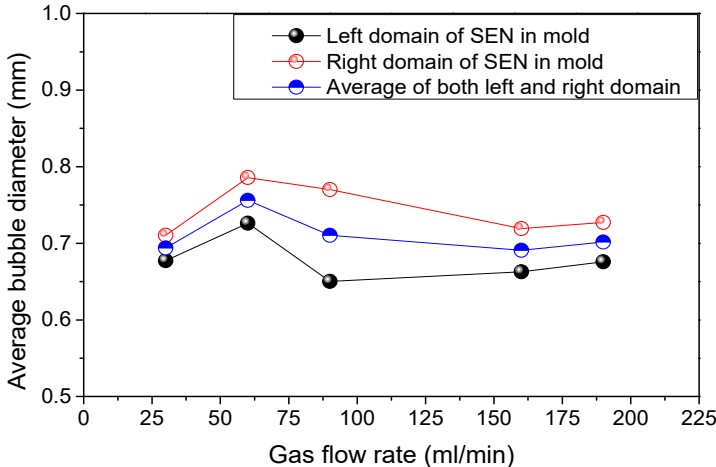

**Figure 14.** Effect of gas flow rate on the average diameter of bubbles.

### 3.4. Study on Gas–Liquid Two-Phase Flow in a Wide Slab Continuous Casting Mold

The fluid flow pattern in the mold will be affected by gas injection. Figure 15 shows four instantaneous snapshot velocity contours and vectors on the thickness center plane of the mold with the gas flow rate of 30 mL·min$^{-1}$. At 0.00 s, the density of the gas–liquid mixed phase is much smaller than that of the liquid phase alone, as shown in Figure 15a. The liquid phase with bubbles is ejected from the nozzle and moved to the liquid surface by buoyancy, and only a small portion of the liquid phase without air bubbles enters the lower circulation. The velocity of the lower circulation is significantly weakened, the vortex center of the upper circulation moves toward the nozzle, and the liquid velocity in the vertical direction of the liquid surface increases significantly. At 2.60 s, as shown in Figure 15b, the moment from the gate on the right side into the mold of the bubble is greater than the left, so the nozzle on the left side of the mold flow is similar to single-phase flow. That is, the flow of the impact to the narrow surface is divided into upwelling and down flow, but compared with single-phase flow, the upper circulation vortex center is closer to the inlet. The right side of the nozzle is gas–liquid two-phase flow, which is similar to 0.00 s. At 6.24 s, as shown in Figure 15c, the number of bubbles entering the mold from the left side of the nozzle increased gradually, but it was still much less than that from the right side. At 26.52 s, as shown in Figure 15d, the bubbles entered the mold from the left and right sides of the nozzle more evenly, and the flow pattern in the mold returned to the condition of 0.00 s. Through the analysis of the four flow patterns, it can be found that the movement of bubbles from the nozzle into the mold is random, in a particular transient state it is not an average equal amount from the two outlets into the mold, which easily causes asymmetry of the instantaneous flow field, affecting the stability of the slab quality.

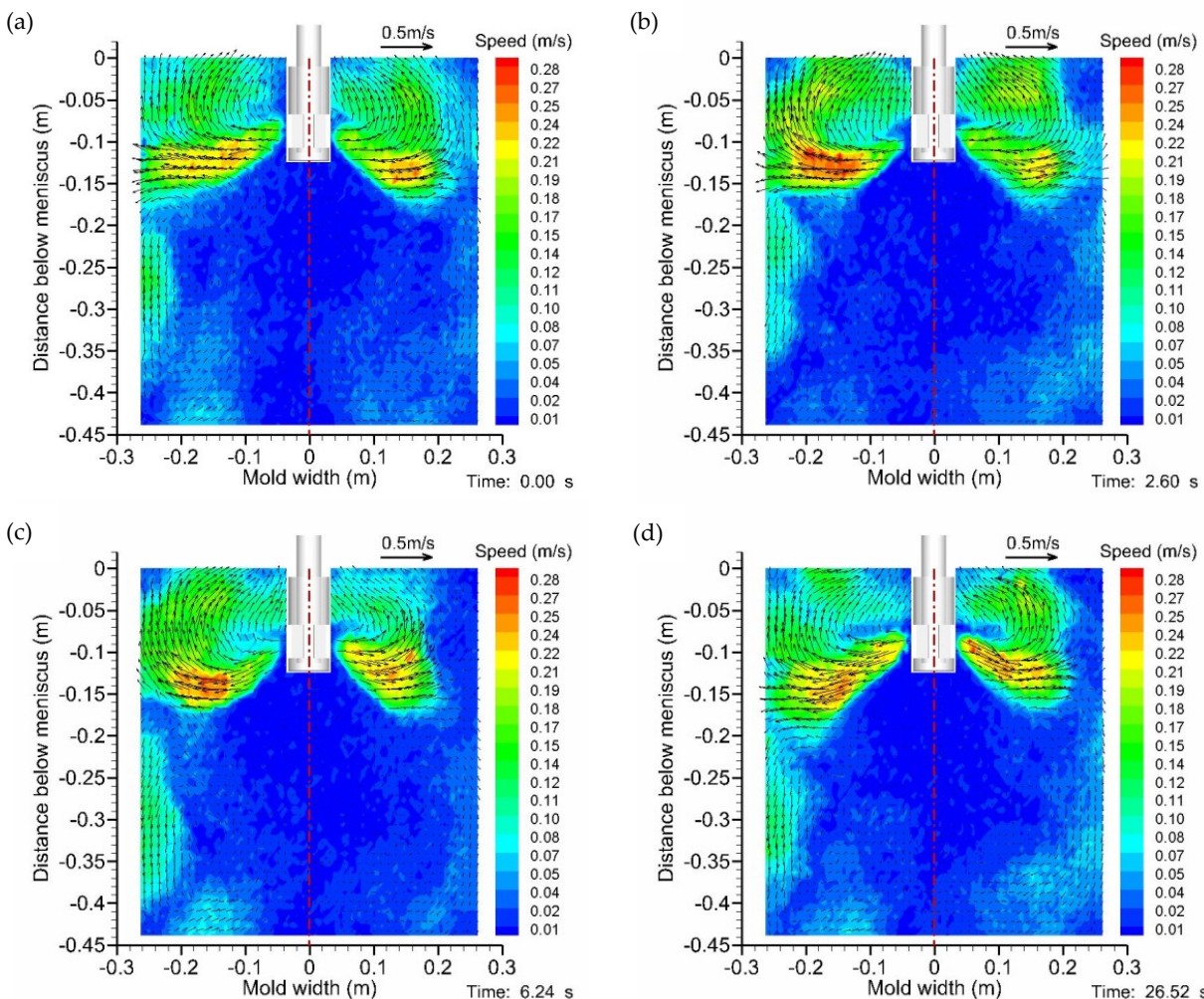

**Figure 15.** The instantaneous velocity field of gas–liquid two-phase flow in the wide slab continuous casting mold changes with time: (**a**) 0.00 s, (**b**) 2.60 s, (**c**) 6.24 s, (**d**) 26.52 s.

By comparing Figure 16a,b, the total amount of bubbles entering the mold from the left and right sides of the nozzle tends to be half of the total over a long enough period of time. From the time-uniform flow of gas–liquid two-phase turbulence, it can be seen that the flow pattern in the mold is still double circulation after blowing. However, compared with the single-phase flow, the vortex center of the upper circulation is closer to the nozzle and the vortex center of the lower circulation is closer to the narrow surface. It can be found that the flow velocity becomes weaker at the narrow surface after blowing, which is the result of the shift of the upper circulation vortex center to the nozzle. In actual production, as the cooling intensity at the meniscus is already high, if the steel flow rate slows down it can lead to a lack of heat supply at the meniscus, easy to get too cold on the meniscus, mold slag being difficult to melt, and slag ring and other phenomena affecting production.

In order to study the change in flow velocity in molds with the increase in gas flow rate under the condition of constant water model casting speed, a point P was selected at 200 mm below the liquid level at the central position of the mold, as shown in Figure 17a. PIV was used to measure the duration of the mold parallel to the central plane of the broad plane 52 under different gas flow rates, and the instantaneous velocity field under each group of blowing volume to find its time-averaged velocity field, and then the time-averaged velocity output of point P under different blowing volumes. The results are shown in Figure 17b. It can be seen that with the increase in gas flow, the average velocity of P point increases gradually, although the casting speed remains unchanged. This indicates that blowing enhances the agitation of flow field in the mold. After the gas–liquid two-

phase flow is ejected from the nozzle, as the bubbles float up, the liquid phase with fewer bubbles flows down to form a lower recirculation zone. As the water model casting speed is constant, the lower return flow meets at the bottom of the mold to form an upward flow and the whole process increases as the gas flow rate increases.

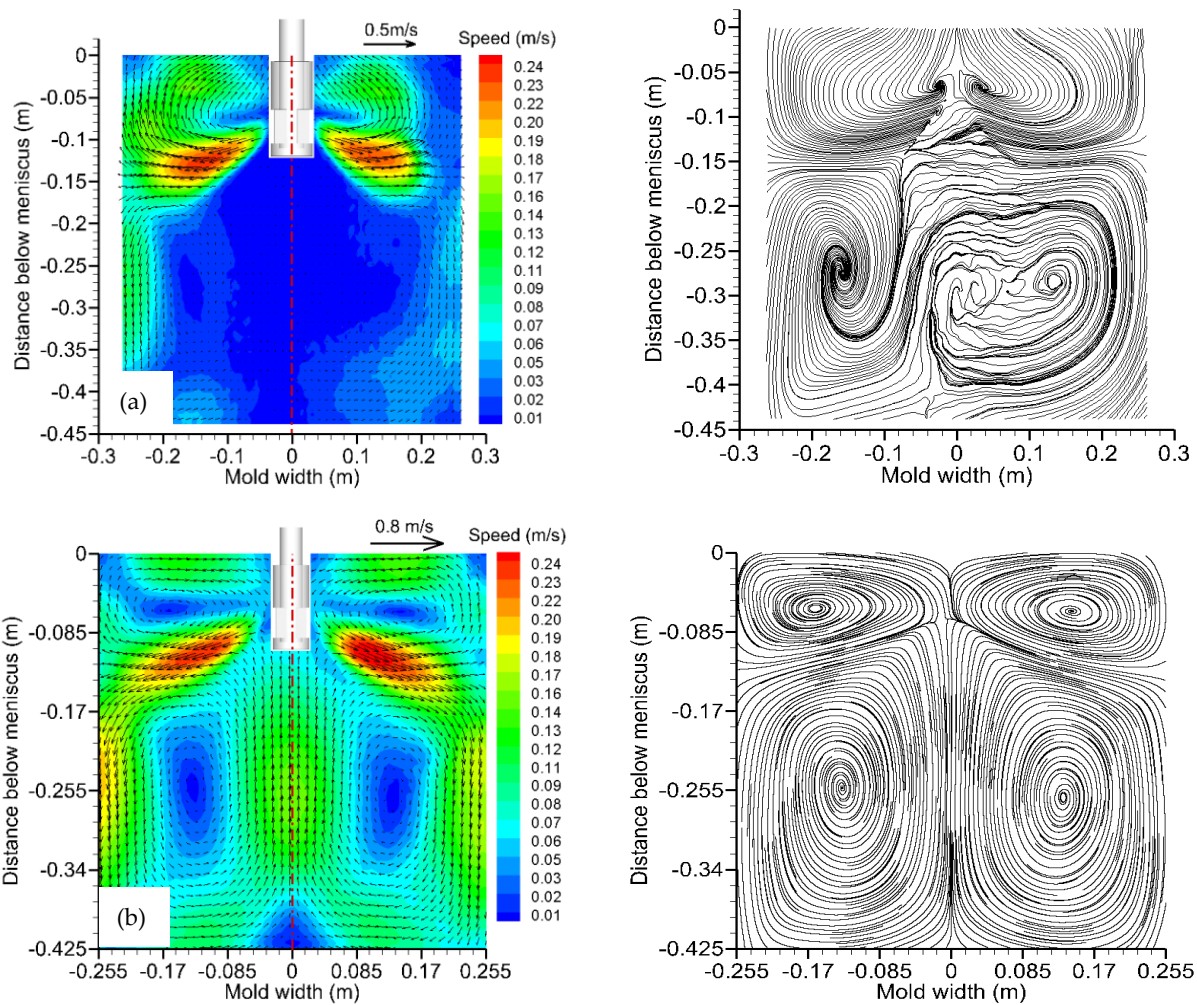

**Figure 16.** Time-averaged velocity contours, vectors, and streamlines with different gas flow rates. (**a**) The blowing volume is 30 mL·min$^{-1}$, (**b**) no blowing.

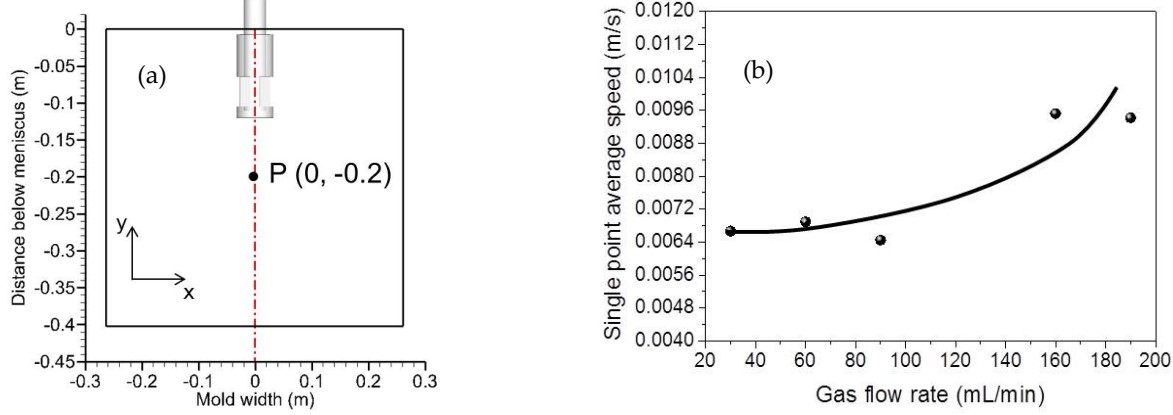

**Figure 17.** The relationship between the flow rate at a specific point P in the wide slab continuous casting mold and the blowing rate. (**a**) The position of the specific point P and (**b**) the effect of blowing rate on the flow rate at point P.

In order to further analyze the characteristics of gas–liquid two-phase flow in a wide slab casting mold, the horizontal surface flow velocity of liquid level and vertical flow velocity of a narrow surface were compared in the time-averaged velocity field of the mold under different blowing gas levels. Figure 18 shows the schematic diagram of the output position of the liquid level and narrow surface flow velocity.

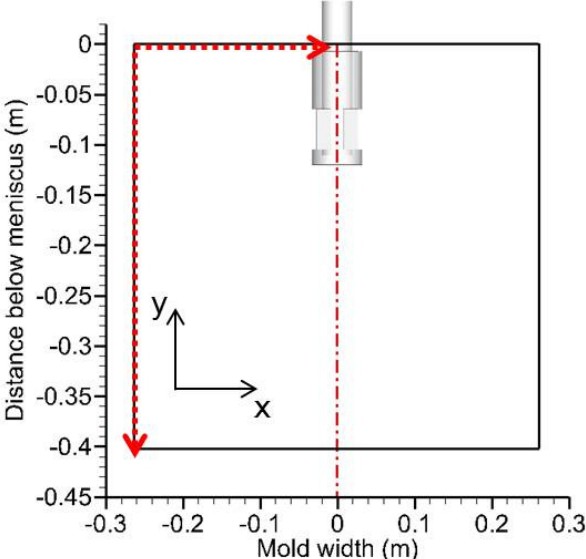

**Figure 18.** Schematic diagram of the output position of the liquid surface and the narrow surface.

Figure 19 shows the relationship between the gas flow rate and the horizontal surface flow velocity. It can be seen that blowing has a suppressive effect on the liquid surface flow rate. With the increase in the gas flow rate, the maximum velocity gradually decreases and tends to approach the nozzle. When gas flow rate is 30, 60, 160, and 190 mL·min$^{-1}$, respectively, the maximum flow velocities in the horizontal direction are 0.105, 0.08, 0.055, and 0.045 m·s$^{-1}$, respectively, and the nozzle immersion depth is 150, 145, 112, and 130 mm, respectively. This result shows that the larger the gas flow rate is, the larger the bubble size is, the larger the buoyancy is, and the shorter the horizontal movement distance after the bubble is ejected from the nozzle. In other words, the closer the bubble escapes from the liquid surface, the closer it is to the nozzle, and the closer the vortex center of the upper circulation is to the nozzle. Figure 20 shows the relationship between the gas flow rate and the vertical flow velocity of the narrow plane. It can be seen that with the increase in the gas flow rate, the vertical flow velocity of the narrow plane does not change significantly, and the change in the gas flow rate does not affect the position of the jet flow impacting the narrow surface.

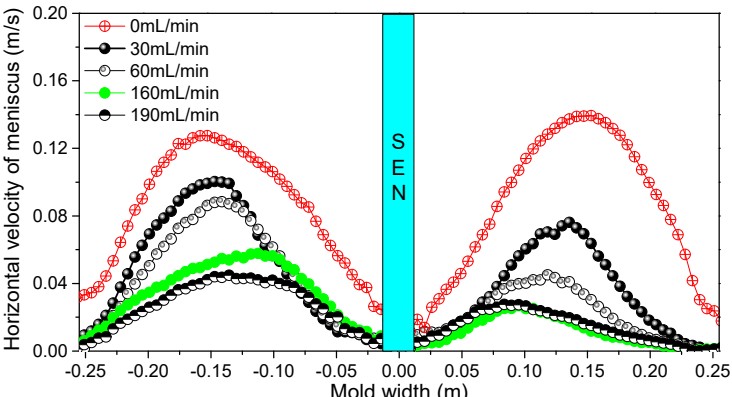

**Figure 19.** Effect of gas flow rate on the surface horizontal velocity.

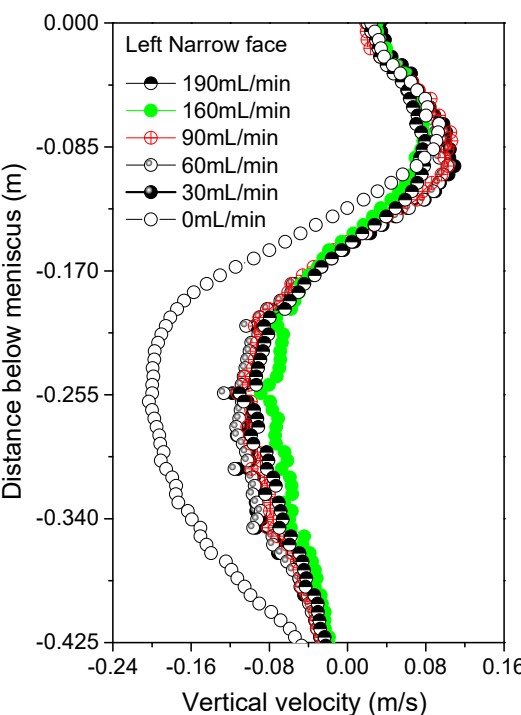

**Figure 20.** The relationship between the gas flow rate and the vertical velocity of the narrow surface.

Figure 21 shows the output positions of gas–liquid two-phase flow rate and velocity at different distances from the narrow surface. Starting from the narrow surface, the interval is 18–72 mm from the narrow surface. Figure 22 shows the comparison of the hourly average velocity magnitudes on the vertical lines of 0, 18, 36, 54, and 72 mm from the narrow surface for a gas flow rate of 30 mL·min$^{-1}$. It can be seen that the liquid surface flow rate at the narrow side is the lowest and increases as the distance from the narrow side increases.

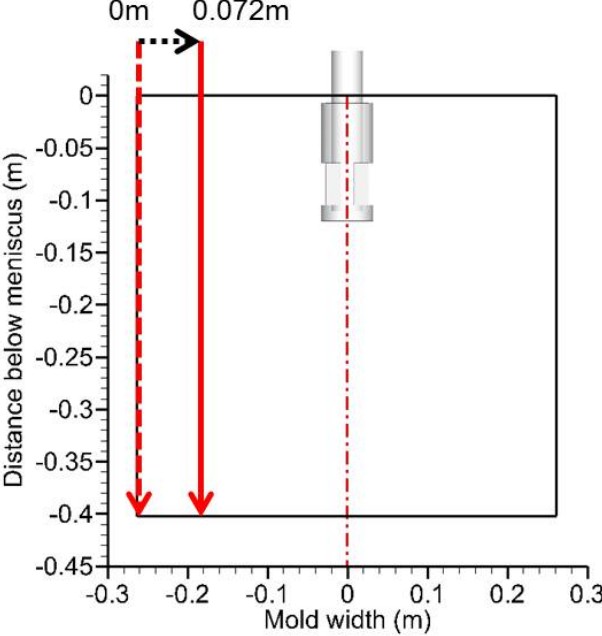

**Figure 21.** Schematic diagram of the output position of the velocity and velocity at different distances from the narrow surface.

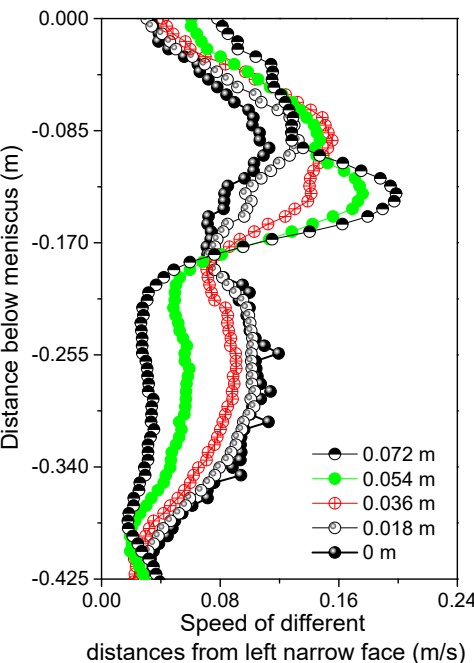

**Figure 22.** Comparison of flow velocity and velocity at different distances from the narrow surface (gas flow rate is 30 mL·min$^{-1}$).

## 4. Conclusions

(1) When oil is added, the speed of the liquid surface around the nozzle decreases, and the higher the viscosity of oil is, the higher the critical casting speed (maximum casting speed of no slag entrapment) is, and the lower the critical immersion depth of SEN is.

(2) Blowing has an inhibiting effect on the liquid surface velocity. With the increase in the gas flow rate, the maximum value of the flow rate in the horizontal direction of the liquid surface gradually decreases, and the location where the maximum flow rate occurs has a tendency to move closer to the nozzle. However, the change in flow velocity in the vertical direction of the narrow surface is not obvious, and the change in gas flow rate will not affect the position of the jet impacting the narrow surface.

(3) The bubble size and the number of bubbles within 1 mm in diameter increase with the increase in gas flow rate, but the bubble size does not always increase. The larger the size of the bubble, the greater the buoyancy force, the bubble escapes from the liquid surface closer to the SEN, the vortex center of the upper circulation is also closer to the water mouth, the vortex center of the lower circulation is closer to the narrow surface, and the liquid surface velocity decreases at the meniscus.

(4) The bubbles enter the mold from the nozzle with randomness, easily causing asymmetry of the transient flow field. The agitation of the flow field in the mold will be enhanced by increasing the gas flow rate, but the flow state in the mold is still double circulation.

**Author Contributions:** Conceptualization, L.R.; methodology, L.R.; validation, L.R., W.L., H.L. and J.Y.; formal analysis, L.R.; investigation, L.R.; resources, L.R. and J.Y.; data curation, L.R.; writing—original draft preparation, W.L.; writing—review and editing, L.R.; visualization, L.R.; supervision, L.R.; project administration, L.R.; funding acquisition, L.R. All authors have read and agreed to the published version of the manuscript.

**Funding:** This research was funded by the National Natural Science Foundation of China grant number 51804171 and 51774190. And The APC was funded by 51804171.

**Institutional Review Board Statement:** Not applicable.

**Informed Consent Statement:** Not applicable.

**Data Availability Statement:** The raw/processed data required to reproduce these findings cannot be shared at this time as the data also forms part of an ongoing study.

**Acknowledgments:** The authors are grateful for support from the National Natural Science Foundation of China (Grant No. 51804171 and No. 51774190).

**Conflicts of Interest:** The authors declare no conflict of interest.

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
