# Peer review of "Study on Multiphase Flow in a Wide-Width Continuous Casting Mold"

_processes, doi:10.3390/pr10071269_

Round 1

Reviewer 1 Report

After careful reading of the article, the manuscript identification processes-1770118, Study on Multiphase Flow in a Wide-Width Continuous Casting Mold, the following minor remarks arise:

1.  Is the superscript formatting of the cited literature incorrect? 2.  To the table on page 2, you would need to add a title and reference in the text. 3.  The part descriptions in fig. 2 are too small. 4.  Fig. 3 is incorrectly dimensioned and the font is illegible. 5.  Table 1 (whether table I, 2 or II), page 3 should be aligned to the text (from the left), like table2, page 4, and may fit the width of the text. 6.  The descriptions of the parts in fig.4 are too small, please enlarge the font. 7.  Line 111 high, it should be High ……… 8.  Are fig. 6 and all others not left-aligned? 9.  Please check the capitalization and formatting in the caption fig. 9, 10, 11 and 12. 10.Line 300, 301, please pay attention to moving the text, units to the next line. Also elsewhere in the text.   The article is interesting, quite difficult, with unexpected results, but it contains accurate conclusions and observations.
The purpose and scope of the research is clear and the presentation of the results is short and specific.

Reviewer 2 Report

The bubble-water-oil system are used to study multiphase flow in continuous casting mold. The effect of slag on fluid flow, slag entrapment, bubble behavior, and flow velocity were studied.

Some comments:

1) Technically comments:

Could protective slag be mold slag in all the manuscript?

Use English in figure 3. Also the detailed parameter is not clear. Please redraw the figure.

Could the flow rate in page 2 table 1 be the velocity, otherwise the unit should be m3/s.

Figure 8 might be named as oil entrapment photos, since the K-H instability was not studied herein. The K-H might be derived systematically.

Please refer the casting speed as water model casting speed in figure 9 and in many other places, otherwise the readers may thought it as casting speed in plant.

What does critical drawing rate mean in page 6 line 158?

Figure 11 could be snapshot of bubble distribution figure, a series of figure at different time might be named as trajectory.

Page 13 line 264, the billet might be replaced by slab.

What does water mouth mean in conclusion 3? This was not mentioned in the manuscript.

2) Language part:

Page 2 line 66, the law maybe replaced by mechanism.

Page 2 line 72, rewrite the sentence ‘which..’

Page 2 line 74, pulling speed maybe casting speed.

Please rewrite the sentence in page 4 line 106-107, the experiment will be…set to..

Also in page 4 line 127, “the right same as the left side”

Tittle of section 3.2 might be slag entrapment, not slagging.

Please rewrite the sentence in page 13 line 275-277.

Please rewrite the conclusion 1. The critical may refers to the slag entrapment etc.

3) Format:

Use reference numbers [3] in 2nd and 3rd paragraphs in page 1.

Table 1 was missing in page 2.
